# The Impact of Diabetes Mellitus on Cardiovascular Risk Onset in Children and Adolescents

**DOI:** 10.3390/ijms21144928

**Published:** 2020-07-12

**Authors:** Ida Pastore, Andrea Mario Bolla, Laura Montefusco, Maria Elena Lunati, Antonio Rossi, Emma Assi, Gian Vincenzo Zuccotti, Paolo Fiorina

**Affiliations:** 1Division of Endocrinology, ASST Fatebenefratelli-Sacco, 20157 Milan, Italy; ida.pastore@asst-fbf-sacco.it (I.P.); andrea.bolla@asst-fbf-sacco.it (A.M.B.); montefusco.laura@asst-fbf-sacco.it (L.M.); mariaelena.lunati@asst-fbf-sacco.it (M.E.L.); rossi.antonio@asst-fbf-sacco.it (A.R.); 2International Center for T1D, Pediatric Clinical Research Center Romeo ed Enrica Invernizzi, Department of Biomedical and Clinical Science L. Sacco, University of Milan, 20157 Milan, Italy; emma.assi@unimi.it; 3Pediatric Clinical Research Center Romeo ed Enrica Invernizzi, DIBIC, Università di Milano and Department of Pediatrics, Buzzi Children’s Hospital, 20157 Milan, Italy; gianvincenzo.zuccotti@unimi.it; 4Nephrology Division, Boston Children’s Hospital, Harvard Medical School, Boston, MA 02115, USA

**Keywords:** diabetes mellitus, children, adolescents, cardiovascular risk

## Abstract

The prevalence of diabetes mellitus is rising among children and adolescents worldwide. Cardiovascular diseases are the main cause of morbidity and mortality in diabetic patients. We review the impact of diabetes on establishing, during childhood and adolescence, the premises for cardiovascular diseases later in life. Interestingly, it seems that hyperglycemia is not the only factor that establishes an increased cardiovascular risk in adolescence. Other factors have been recognized to play a role in triggering the onset of latent cardiovascular diseases in the pediatric population. Among these cardiovascular risk factors, some are modifiable: glucose variability, hypoglycemia, obesity, insulin resistance, waist circumference, hypertension, dyslipidemia, smoking alcohol, microalbuminuria and smoking. Others are unmodifiable, such as diabetes duration and family history. Among the etiological factors, subclinical endothelial dysfunction represents one of the earliest key players of atherosclerosis and it can be detected during early ages in patients with diabetes. A better assessment of cardiovascular risk in pediatric population still represents a challenge for clinicians, and thus further efforts are required to properly identify and treat pediatric patients who may suffer from cardiovascular disease later in early adulthood.

## 1. Introduction

The prevalence of the two most common forms of diabetes, type 1 (T1D) and type 2 (T2D) is increasing worldwide, even in the pediatric population, rapidly becoming an urgent public health problem [1,2,3]. The International Diabetes Federation has estimated that T1D affects more than 1,100,000 children and adolescents, with an annual incidence of about 128,900 worldwide [1]. Until the early 1990s, T2D was believed to be almost an exclusive condition of adulthood, but its prevalence in adolescents is increasing in many countries [1]. Even if the estimates are not as precise as for T1D, in the United States, 20,262 adolescents are diagnosed with T2D, and by 2050, a fourfold increase will occur [3,4]. In addition, prediabetes affects nearly 5% of children aged between 6 and 10 years [5]. These growing trends are largely due to the spread of a more western lifestyle. Physical inactivity, increased calorie intake and reduced energy expenditure all facilitate overweight and obesity onset in children [6]. These conditions affect almost all children with T2D, and are common also in the pediatric population, with T1D having a prevalence of up to 34% [7,8,9]. A recent study was conducted in a population of 708 children who were positive for one of the circulating diabetes autoantibodies, but were not diabetic. The study revealed that the children who tested negative for HLA haplotypes predisposing to T1D and had an elevated body mass index (BMI) progressed to multiple autoantibodies positivity and had an increased risk of developing T1D [10]. Younger patients with T2D showed an impaired insulin secretion and detectable circulating autoantibodies [11,12]. Moreover, children and adolescents with T2D, when compared to adults with T2D, showed a higher and faster loss of beta-cells activity, which makes achieving optimal metabolic compensation more difficult [13,14,15]. 

Childhood and adolescence are crucial stages of life for the onset of cardiovascular (CV) risk factors [16]. There is a large body of evidence that suggests that the prevalence of CV risk factors among diabetic children and adolescents is high and, in most cases, these factors are present already at the time of diagnosis [8]. Diabetes mellitus is associated with a two-fold increase in the risk for cardiovascular disease, with a premature CV mortality and a four-fold increase in mortality for all-cause in young [17,18,19]. Moreover, the coexistence of obesity further increases the risk for specific- and all-cause mortality [20]. Previous studies in T1D and T2D, although not conclusive, suggested an increased risk for ischemic heart, macrovascular diseases and death, particularly in patients with T2D diagnosed between 15 and 30 years of age [21]. A longitudinal cohort of 6.840 patients with T1D, and 1.518 patients with T2D, showed an excess mortality in patients with T2D aged between 15 to 19 when compared to the general population [22]. Moreover, in patients with T1D, CV outcomes and mortality inversely correlate with age at diabetes onset; thus suggesting that the earlier the onset, the greater the risk [18]. The prevalence of two or more CV disease risk factors is higher in younger patients with T2D (92%) than those with T1D (14%), and is increased by 1.4% on an annual basis, over the ten years of the study period, in patients with T2D, but not in those with T1D [23,24]. In line with the aforementioned study, Dabalea et al. demonstrated that 72% of 272 young patients with T2D, contrasting only 32% of 1746 young patients with T1D, developed diabetic complications [25]. The role of hyperglycemia in establishing cardiovascular risk in the short-term might not be as clinically evident in children and adolescents with diabetes when compared to adults with diabetes, partially due to the greater potential regenerative capacity and to the higher number of circulating endothelial progenitor cells in the young [26]. However, in the long-term, evidence suggests a more detrimental role of diabetes when present at a younger age in making patients more vulnerable to cardiovascular CV risks later in life [18]. 

## 2. CV Risk Factors in Children and Adolescents with Diabetes

There are several old and new CV risk factors that appear to be relevant for pediatric patients with diabetes and that can be targeted. We classify them as modifiable and unmodifiable factors (Table 1). Among those unmodifiable factors are the age at onset and the disease duration [18,21,24,27]. A large longitudinal study comprising 27,195 patients with T1D and 135,178 controls, proved that mortality for CV disease and for all-cause inversely correlates with the onset age of T1D [18]. Moreover, patients diagnosed before 10 years of age showed a reduced life expectancy of nearly 18 years for women and 14 years for men [18]. Similarly, an early onset of T2D worsens CV risk, and younger patients with T2D are exposed to higher rates of diabetic micro- and macro-vascular complications [21,25]. As the duration of the disease increases, the prevalence of CV risk factors, CV disease death, myocardial infarction, revascularization, angina and stroke rises as well [24,27]. Contrary to what has been reported for the adult population whereby the female gender is associated with a lower CV risk in pre-menopausal period, data from recent literature reported similar CV risk rates in males and females [27,28,29]. Several modifiable CV risk factors are related to anthropometric measurements and metabolic control [30]. Increased BMI and waist circumference are described in children and adolescents with diabetes, while elevated levels of HbA1c are positively correlated with increased macrovascular complications [1,24,27,31]. Recently, increased glucose variability, which refers to the number and the amplitude of blood glucose fluctuations, more than the mean glycemia or the HbA1c, has been suggested to be a novel CV risk factor [32,33]. A reduction in glucose variability in young patients, with T1D by using continuous subcutaneous insulin infusion and real time continuous glucose monitoring, ameliorates endothelial function by increasing the flow-mediated dilatation of the brachial artery [34]. Another modifiable risk factor is hypoglycemia. Indeed, Fahrmann et al. showed that severe hypoglycemia positively correlates, regardless of age, with coronary artery calcification in the sub-cohort of patients enrolled in DCCT/EDIC study [35]. Insulin resistance is common to both T1D and T2D, although its pathophysiology seems to be different in youth with T1D, as compared to those with T2D [36]. The simultaneous presence of insulin resistance and T1D has been defined as “hybrid or double diabetes” and patients with T1D and insulin resistance own a more elevated CV risk among patients with T1D [37]. The control of other abnormalities associated with diabetes is of paramount importance. For instance, the importance of assessing blood pressure is supported by the observation that children with T1D with elevated systolic blood pressure had increased carotid intima-media thickness [38]. The prevalence of microalbuminuria, another modifiable risk factor, increases proportionally in youth with T2D, together with the disease duration [39], while in youth with T1D, this mainly correlates with elevated arterial stiffness [40]. In youth with diabetes, a more atherogenic lipid profile directly correlates with glycated hemoglobin, fasting glucose, age, disease duration and the presence of insulin resistance. Many of the aforementioned factors may be associated with a reduced nitric oxide availability [30,41,42,43,44]. Finally, the role of smoking and alcohol as CV risk factors in adulthood is well known, even if their long-term effects in adolescents have not been clarified yet [45].

## 3. Inflammation in Children and Adolescents with Diabetes 

An early appearance of a pro-inflammatory state may be a key player in conditioning the onset of CV disease later in life [46]. Central obesity and physical inactivity are both associated with a pro-inflammatory state and are very common in young patients with obesity, with T2D and with T1D [9,11,13,42,47]. Obesity and T2D are characterized by a state of systemic chronic low-grade inflammation, that triggers a vicious circle involving insulin resistance, oxidative stress and endothelial dysfunction and lays the basis for early and accelerated atherosclerosis [46]. A low-grade inflammation is also observed in lean children with T1D, while increased levels of pro-inflammatory cytokines are described, either in T1D or T2D [48,49]. Continuous systemic chronic inflammation from childhood accelerates plaque formation and contributes to its growth [48]. Nearly 700 adolescents aged between 10 and 17 years who had a recent diagnosis of T2D were studied, and a detrimental inflammatory profile that worsened over time was shown [50]. Indeed, the inflammatory state was only partially reverted by pharmacological therapy [50]. In children and adolescents with diabetes, pro-inflammatory abnormalities may play a prognostic role for the development of diabetic complications and may represent novel pharmacological targets [51]. Previous studies, although not conclusive, demonstrated increased levels of C-reactive protein, interleukin-6, tumor necrosis factor-α, leptin, and decreased levels of adiponectin in children and adolescents with diabetes [52,53,54,55,56,57]. Interestingly, pro-inflammatory marker levels seem to also be elevated in lean adolescents with diabetes and good glycemic control [58,59]. Interestingly, the role of diet in modifying the inflammatory profile in children with diabetes is still debated; while in children with obesity, it only seems to be more evident [60,61]. More recently, a putative role for microbiome in the activation of gut and systemic inflammation was suggested in children and adolescents with diabetes [62]. This is of particular interest, especially in children and adolescents in which a pro-inflammatory state may persist over time, and may exert a detrimental effect on the CV system [46]. Interventional studies on the effect of prebiotics on gut microbiota are needed to elucidate their role in improving inflammatory state, insulin resistance and glycemic control.

## 4. Endothelial Dysfunction in Children and Adolescents with Diabetes

Even if not entirely well elucidated in their mechanisms, vascular complications are the results of direct action of hyperglycemia, but also of abnormalities of other mediators such as proinflammatory cytokines, growth factors, advanced glycation end-products (AGEs) and cell adhesion molecules [63,64]. Endothelial injury may represent the earlier phenomenon of vascular dysfunction, and hyperglycemia is probably the main driver of endothelial injury in children [65]. Hyperglycemia increases the production of reactive oxygen species and of AGEs, activates the protein kinase C, hexosamine and polyol pathways and affects the function of endothelial progenitor cells [51,66,67]. A growing number of studies showed increased inflammation and endothelial dysfunction in children and adolescents with diabetes, which was then associated with an increased CV risk (Table 2).

Children and adolescents with T1D showed elevated levels of the highly sensitive C-reactive protein when compared to healthy subjects [53,68]. In children and adolescents with diabetes, an increased level of serum intercellular adhesion molecule-1 and vascular cell adhesion molecule-1 was observed [47,55,69,70,71]. Levels of E-selectin are augmented in children and adolescents with T1D and are associated with systolic and diastolic blood pressure abnormalities [69,72]. Furthermore, high levels of circulating endothelial cells in children and adolescents with T1D positively correlated with HbA1c levels [53]. Increased levels of tumor necrosis factor-α were observed in young patients with T1D and T2D, and high interleukin-6 levels were observed in those with T1D [54,55,70,73]. Unfortunately, the clinical significance of these biomarkers must still be proven, and their use in epidemiological studies still needs to be explored. In this view, the early detection of endothelial dysfunction in children and adolescents with diabetes might be useful to prevent, or at least delay, cardiovascular complications [74].

## 5. Cardiac Dysfunction in Children and Adolescents with Diabetes 

Diabetic cardiomyopathy comprises both functional and structural changes of the myocardium, regardless of hypertension, ischemic or valvular heart disease, and it can occur either as a systolic or diastolic dysfunction [75]. Hyperglycemia leads to the increased tissue deposition of AGEs that crosslink with proteins of the extracellular matrix, resulting in fibrosis and myocardial tissue remodeling [76]. The role of hyperglycemia in initiating and maintaining left ventricular diastolic dysfunction is supported by the observation that, in patients undergoing combined kidney and pancreas transplantation, no longer requiring insulin administration, a reversal of diastolic dysfunction was observed [77]. Endothelial dysfunction and vascular stiffness precede diastolic dysfunction, and their early recognition and treatment may prevent or delay diastolic dysfunction [78]. As left ventricular diastolic dysfunction represents the earliest manifestation of the diabetic cardiomyopathy that anticipates systolic dysfunction, several studies revealed diastolic dysfunction still in the presence of a normal left ventricular ejection fraction [77,79]. Children and adolescents with diabetes manifested subclinical ventricular abnormalities, when studied with both conventional or novel tissue Doppler echocardiography, and compared with healthy controls [80,81,82]. Moreover, concentric left ventricular hypertrophy is common among adolescents with diabetes, especially in those with obesity, identified by the increase of left ventricular wall thickness over time [82].

Children and adolescents with a recent onset of T1D showed the same cardiac function of healthy peers [83]. In a prospective cross-sectional study, it has been demonstrated that effectively controlled children and adolescents with long-term T1D showed diastolic dysfunction and subclinical systolic dysfunction, regardless of a normal left ventricular ejection fraction, and both of these abnormalities are associated with disease duration [81]. Cardiac functional abnormalities are associated with BMI in children and adolescents with and without diabetes, suggesting that the control of obesity is mandatory to prevent cardiac disease, especially in those with diabetes in which the progression of cardiac abnormalities seems accelerated [82,84,85]. In children with T1D, cardiac dysfunction seems to be influenced by disease duration, poor glycemic control, microalbuminuria, retinopathy and increased blood pressure [83,86,87]. Left ventricular performance has also been studied during physical activity sessions in adolescents with and without diabetes and, as expected, those with T1D or T2D showed a lower exercise tolerance compared to healthy controls [88,89]. Interestingly, a structured physical activity program over time is able to ameliorate physical training tolerance in adolescents with and without diabetes, but the level of cardiac performance remains lower in those with diabetes [89,90]. Nevertheless, these observations confirm the essential role of physical exercise in a necessarily modified lifestyle, adapted for the prevention of CV disease, especially in the high-risk CV population [89]. 

## 6. Vascular Tests to Unveil Early CV Risk in Children and Adolescents with Diabetes

Several surrogate indices are used in clinical practice to assess early vascular abnormalities (Table 3**).** Young patients with T1D showed delayed or reduced brachial artery flow mediated dilatation (FMD) reactivity and higher carotid femoral pulse wave velocity (PWV), as compared to healthy controls [53,54,59,68,78,91,92,93,94,95,96]. On the other hand, Bradley et al. demonstrated that carotid-radial PWV, but not carotid-femoral PWV, is higher in adolescents with T1D than in healthy controls [93]. In young patients with T1D, arterial stiffness is positively associated with glycated hemoglobin levels, disease duration and insulin resistance [97]. When it comes to carotid intima-media thickness (cIMT), an early marker of atherosclerosis, the SEARCH CVD study by analyzing data from 298 adolescents with diabetes, showed that BMI is a strong predictor of cIMT over time [45], and that cIMT increases in young patients with the worst glycometabolic control [98,99]. The mean cIMT correlates positively with hypertension, retinopathy, microalbuminuria, and in males with HDL cholesterol too [100]. 

## 7. Early Treatment of Children and Adolescents with Diabetes

In pediatric patients with diabetes, CV risk begins early and grows over time, and thus maintaining an optimal glycometabolic control in the long term is mandatory [101,102]. In the management of children and adolescents with diabetes, a pivotal role is played by health education [101]. The identification of a nutritional educational plan, with the aim of managing the intake of nutrients is essential [101]. Regular physical activity improves glycemic control, insulin sensitivity, lipid profile, body composition, wellbeing and cardiovascular health [103]. Therapeutic strategies used in the treatment of young patients with T1D and T2D partially differ from those used in adults [101]. Indeed, insulin therapy is recommended in children and adolescents with T1D, while metformin, insulin and liraglutide are approved for clinical use in children and adolescents with T2D [101]. Intensive insulin therapy allows one to achieve glycemic control, while it can facilitate weight gain, central adiposity, rises in blood pressure, and a more atherogenic lipoprotein profile [104,105]. In young patients with T2D, a more aggressive treatment with metformin and rosiglitazone results in a better durability, with beneficial consequences also on the development of micro- and macro- complications [106,107]. Recently, a secondary analysis of the same population demonstrated that metformin ameliorates lipoprotein profile, glycemic control, blood pressure and BMI, suggesting the role of metformin in improving CV parameters [108]. Data from a pediatric diabetes consortium registry analyzing young patients with T2D demonstrated the durability of metformin monotherapy in those with lower HbA1c and a more recent onset of the disease [109]. A growing number of studies showed that metformin enhances insulin sensitivity and reduces insulin dose in youth with T1D, but also improves vascular markers [105,110].

The sodium glucose co-transporter-2 inhibitors (SGLT2i) and glucagon-like peptide-1 receptor agonists (GLP-1RA) are two classes of anti-diabetic drugs that have demonstrated to exert cardioprotective effects in multiple cardiovascular and renal outcomes trials in adults with T2D, on primary and secondary prevention [111,112,113,114]. Moreover, their effectiveness and safety are also being tested in the treatment of patients with T1D [115,116,117]. Recently, Tamborlane et al. demonstrated the efficacy of liraglutide in children and adolescents with diabetes aged between 10 to 17, but they also observed an increased number of gastrointestinal adverse effects [118].

In line with these observations, novel antidiabetic therapies should be tested as a treatment of young patients, either with T1D or T2D, with the aim not only of optimizing glycemic control, but also of mitigating their CV risk (Table 4). Further studies are needed to evaluate SGLT2i and GLP1RA efficacy and safety in large cohorts of young patients with diabetes. Current guidelines recommend the use of drugs to control hypertension and dyslipidemia after changes in lifestyle in children and adolescents with diabetes [102,119]. However, there is concern about the use of statins and antihypertensive drugs in this population, for the lack of large intervention trials in the young and their teratogenic effect. More recently, regenerative therapy has proven to be a useful and innovative strategy in preventing and treating vascular complications in diabetes. Even further clinical trials are required; these therapies might be of particular interest in the young, who have a higher cellular regenerative capacity than adults [51,120].

## 8. Conclusions

CV disease still remains the leading cause of mortality in diabetes, and the onset of diabetes in pediatric age results in an increased risk for lifelong CV disease. The prevention, or at least the delay, of diabetic complications still represents a challenge for clinicians and caregivers of pediatric patients. If targeted early, cardiovascular complications may be potentially reversible, so interventions should be initiated as soon as possible to avoid the establishment of potentially untreatable CV disease. The CV risk in children and adolescents not only results from the deleterious effects of hyperglycemia, but also can be mediated by others CV risk factors, such as dyslipidemia, hypertension, albuminuria, overweight or obesity. Efforts are needed to better understand the pathophysiology of CV risk in children and adolescents with T1D and T2D, to avoid CV diseases in patients with diabetes. 

## Figures and Tables

**Table 1 ijms-21-04928-t001:** Modifiable and Unmodifiable Cardiovascular Risk Factors in Children and Adolescents with Diabetes.

Modifiable Risk Factors	Unmodifiable Risk Factors
Obesity	Younger age at diabetes onset
Waist circumference	Diabetes duration
Insulin resistance	Family history
Hyperglycemia	
Hypoglycemia	
Glucose variability	
Hypertension	
Microalbuminuria	
Dyslipidemia	
Smoking	
Alchol	

**Table 2 ijms-21-04928-t002:** Studies Evaluating Cardiovascular Biomarkers in Children and Adolescents with Diabetes.

Author	Publication Year	Sample		Age		HbA1c		DM Duration	Peripheral Biomarkers
		DM	HC	DM	HC	DM	HC		
Glowinska [69]	2005	51	27	15.5 ± 3.8	15.2 ± 2.1	NA	NA	NA	↑ sICAM-1, sVCAM, sE-selectin
Schwab [52]	2007	94	40	12.3 (8.5–16.8)	12.3 (7.5–15.2)	7.7 (6.8–10.0)	4.8 (1.2–5-5)	3.8 (1.8–9.8)	↑ hs-CRP, L-selectin, sICAM, vWF, PAP, PAI
Zorena [56]	2013	53	32	12.5 ± 2.9	13.8 ± 3.3	7.6 ± 1.1	3.2 ± 0.8	4.3 ± 2.5	↑ AGEs, TNF-α, VEGF, IL-12
El Samahy [121]	2013	50	50	9.7 ± 3.4	9.8 ± 3.1	NA	NA	4.5 ± 3.5	↑NO
Machnica [55]	2014	52	20	14.0 ± 3.0	13.0 ± 3.0	7.1 ± 0.9	NA	5.0 ± 1.9	↑ sVCAM, TNF-α, IL-6
Eltayeb [53]	2014	30	30	11.1 ± 3.8	9.8 ± 3.5	9.7 ± 2.2	4.9 ± 0.4	3.9 ± 0.6	↑ hsCRP, CECs ↓ vitamin C
Aburawi 1 [70]	2016	79	47	18.6 ± 4.8	17.5 ± 4.6	9.4 ± 2.1	NA	6.8 ± 4.1	↑ sICAM-1, sVCAM
Aburawi 2 [70]	2016	55	47	23.3 ± 5.8	17.5 ± 4.6	7.8 ± 2.5	NA	4.3 ± 3.1	↑ sICAM-1, sVCAM
El-Asrar 1 [122]	2016	21	30	11.5 ± 3.1	10.7 ± 3.2	7.4 ± 0.9	NA	8.0 ± 1.6	↑Angiopoietin-2
El-Asrar 2 [122]	2016	39	30	11.4 ± 3.7	10.7 ± 3.2	9.1 ± 1.3	NA	8.3 ± 1.8	↑Angiopoietin-2
Sochett [123]	2017	51	59	14.8 (10.9–16.8)	13.9 (10.0–17.0)	9.0 ± 1.0	5.0 ± 0.0	6.7 (2.0–16.8)	↑ EGF, PDGF-BB, sCD40L, PDGF-AA, GRO
Rostampour [124]	2017	29	29	11.7 ± 1.9	10.7 ± 2.0	NA	NA	NA	↑ sICAM-1
Fathollahi [71]	2018	48	39	24.2 ± 8.2	28.5 ± 7.2	NA	NA	NA	↑ sICAM-1 = sVCAM ↓ sE-selectin
Karavanaki [125]	2018	56	28	12.0 ± 2.7	12.1 ± 3.3	8.0 ± 1.5	4.1 ± 0.9	5.4 ± 2.8	= OPG, RANKL
Zhang [54]	2019	175	150	12.1 ± 2.5	12.2 ± 1.97	7.8 ± 1.3	4.9 ± 1.6	4.7 ± 2.4	↑ TNF-α, IL-4, hs-CRP, leptin

Data are presented as mean ± SD or median (range). Abbreviations: DM, diabetes mellitus; HC, health control; NA, not available; sICAM-1, serum intercellular adhesion molecule-1; sVCAM-1, serum vascular cell adhesion molecule-1; sE-selectin, serum E-selectin; hs-CRP, high sensitivity-C reactive protein; vWF, von Willebrand factor antigen; PAP, plasmin/ α 2-antiplasmin complex; PAI, plasminogen activator inhibitor; AGEs, advanced glycation end-products; TNF-α, tumor necrosis factor-α; VEGF, vascular endothelial growth factor; IL-12, interleukin-12; NO, nitric oxide; CECs, circulating endothelial cells; EGF, endothelial growth factor; PDGF-BB, platelet-derived growth factor-BB; sCD40L, soluble cluster of differentiation 40 ligand; PDGF-AA, platelet-derived growth factor-AA; GRO, growth regulated oncogene; OPG, osteoprotegerin; RANKL, receptor activator of nuclear factor kappa ligand.

**Table 3 ijms-21-04928-t003:** Studies Evaluating Non-invasive Vascular Test in Children and Adolescents with Diabetes.

Author	Publication Year	Sample		Age		HbA1c		DM Duration	Non-Invasive Vascular Test
		DM	HC	DM	HC	DM	HC		
Urbina [126]	2013	402	206	18.8 ± 3.3	19.2 ± 3.3	8.9 ± 1.8	5.0 ± 0.3	9.8 ± 3.8	↑ bulb cIMT = cIMT, PWV
El Samahy [121]	2013	50	50	9.7 ± 3.4	9.8 ± 3.1	NA	NA	4.5 ± 3.5	↑ cIMT
Eltayeb [53]	2014	30	30	11.1 ± 3.8	9.8 ± 3.5	9.7 ± 2.2	4.9 ± 0.4	3.9 ± 0.6	↑ cIMT ↓ FMD
Ciftel [78]	2014	42	40	13.2 ± 2.6	13.0 ± 2.8	9.0 ± 1.4	NA	6.9 ± 1.7	↑ cIMT ↓ FMD
Pezeshki Rad [127]	2014	40	40	10.6 ± 4.1	10.5 ± 3.2	9.4 ± 2.7	NA	4.2 ± 3.0	↑ cIMT
Atabek [128]	2014	159	100	12.3 ± 4.2	12.2 ± 4.5	9.7 ± 2.5	5.3 ± 1.8	3.8 ± 2.5	↑ cIMT
Shah [40]	2015	402	206	18.8 ± 3.3	19.2 ± 3.3	8.9 ± 1.8	5.0 ± 0.3	9.8 ± 3.8	↑ PWV
Bradley [93]	2016	199	178	14.4 ± 1.6	14.4 ± 2.1	8.5 ± 1.2	5.4 ± 0.2	7.2 ± 3.1	↑ = PWV
El-Asrar 1 [122]	2016	21	30	11.5 ± 3.1	10.7 ± 3.2	7.4 ± 0.9	NA	8.0 ± 1.6	↑ cIMT, aIMT
El-Asrar 2 [122]	2016	39	30	11.4 ± 3.7	10.7 ± 3.2	9.1 ± 1.3	NA	8.3 ± 1.8	↑ cIMT, aIMT
Terlemez [129]	2016	72	77	12.8 ± 3.7	12.3 ± 1.6	8.6 ± 1.9	NA	3.9 ± 2.6	↑ PWV
Nascimento [94]	2017	22	58	8.6 ± 1.7	8.3 ± 1.8	8.8 ± 1.5	5.3 ± 0.2	> 5	= FMD, cIMT
Nascimento [94]	2017	9	58	10.1 ± 1.2	8.3 ± 1.8	9.5 ± 1.7	5.3 ± 0.2	< 5	↓FMD = cIMT
Rostampour [124]	2017	29	29	11.7 ± 1.9	10.7 ± 2.0	NA	NA	NA	↑ cIMT
Pillay [95]	2018	38	28	13.0 ± 2.9	13.9 ± 2.7	8.8(6.6–14)	5.2(4.7–5.7)	5.4 ± 4.6	↓ FMD
Lilje [130]	2018	38	38	13.4 ± 3.4	5.8 ± 4.3	9.7 ± 1.6	NA	5.8 ± 4.3	↑aIMT, fIMT = cIMT
Karavanaki [125]	2018	56	28	12.0 ± 2.7	12.1 ± 3.3	8.0 ± 1.5	4.1 ± 0.9	5.4 ± 2.8	= cIMT
Zhang [54]	2019	175	150	12.1 ± 2.5	12.2 ± 1.9	7.8 ± 1.3	4.9 ± 1.6	4.7 ± 2.4	↑ cIMT, aIMT ↓ FMD
Podgorski [131]	2019	50	50	13.4 ± 3.8	13.1 ± 4.1	7.6 ± 1.2	NA	6.5 ± 3.8	↑ PWV = cIMT
Glackin [132]	2020	57	29	13.9 ± 2.3	15.1 ± 2.2	7.8 ± 1.8	5.3 ± 0.3	5.4 ± 4.1	= cIMT

Data are presented as mean ± SD or median (range). Abbreviations: DM, diabetes mellitus; HC, health control; NA, not available; cIMT, carotid intima media thickness; PWV, pulse wave velocity; FMD, flow mediated dilatation; aIMT, aortic intima media thickness; fIMT, femoral intima media thickness.

**Table 4 ijms-21-04928-t004:** Effects of Hypoglycemic Therapies on Different Outcomes in Children and Adolescents with Diabetes.

Hypoglycemic Therapy	Diabetes Control	Insulin Sensitivity	Blood Pressure	Atherogenic Profile	CV Health	Body Weight
Education	↑	NA	NA	NA	↑	NA
Diet	↑	⟷	NA	↓	⟷	↓
Physical activity	↑	↑	⟷	↓	↑	↓
Insulin	↑	⟷	↑	↑	⟷	↑
Metformin	↑	↑	⟷	⟷	↑	⟷/↓
GLP1RA	↑	NA	⟷	⟷	NA	↓

Abbreviations: CV, cardiovascular; NA, not available.

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
