# Peer review of "The Impact of Diabetes Mellitus on Cardiovascular Risk Onset in Children and Adolescents"

_ijms, 2020, doi:10.3390/ijms21144928_

Round 1
Reviewer 1 Report
The manuscript of Pastore I. et al. is devoted to the impact of diabetes mellitus on the onset of cardiovascular (CV) risk in children and adolescents. The work emphasizes that diabetes mellitus is associated with a two-fold increase in the risk for CV disease and a four-fold increase in mortality for all-cause in young. The review highlights the main factors of CV risk, such as hyperglycemia dyslipidemia, hypertension, albuminuria, overweight, obesity, and others. The review is interesting and relevant. The manuscript may be accepted after minor corrections.
Minor comments
- The authors should revise the abstract to make it more informative and readable. It is necessary to describe in more detail modifiable and unmodifiable risk factors for cardiovascular diseases in children and adolescents with diabetes.
- There are some typographic errors. For example, page 3 - replace “low grade” with low-grade. Please, add a comma: “In a prospective cross-sectional study, it has…” (page 4). The authors are encouraged to proof-read thoroughly the text before resubmission.
Author Response
Responses to Referee 1 comments:
>> The manuscript of Pastore I. et al. is devoted to the impact of diabetes mellitus on the onset of cardiovascular (CV) risk in children and adolescents. The work emphasizes that diabetes mellitus is associated with a two-fold increase in the risk for CV disease and a four-fold increase in mortality for all-cause in young. The review highlights the main factors of CV risk, such as hyperglycemia dyslipidemia, hypertension, albuminuria, overweight, obesity, and others. The review is interesting and relevant. The manuscript may be accepted after minor corrections.
Authors’ response: We thank the Reviewer and appreciate the positive comments.
- The authors should revise the abstract to make it more informative and readable. It is necessary to describe in more detail modifiable and unmodifiable risk factors for cardiovascular diseases in children and adolescents with diabetes.
Authors’ response: We thank the reviewer for the valuable comment and edited the abstract to make the content clearer for the readership and enriched it by introducing the suggested reports. Please, see edited text at page 1 of the manuscript.
- There are some typographic errors. For example, page 3 - replace “low grade” with low-grade. Please, add a comma: “In a prospective cross-sectional study, it has…” (page 4). The authors are encouraged to proof-read thoroughly the text before resubmission.
Authors’ response: We are very sorry for our incorrect writing. The typing errors have been revised carefully, and the whole manuscript has been carefully rechecked.

Reviewer 2 Report
General
It should be clearly underlined if there are important differences in the effect of diabetes on CV risk between adults and children/adolescents as well as in treatment strategy.
The manuscript should be corrected by a native English speaker. Editorial work is also needed.
Minor
What is T1D 2 ??? (page 2)
Table 2 – hsPCR ???- what it is ???
It is not clear what kind of condition is called “glucose variability”
It dose not to be proper to use “ischemic electrocardiogram” in this sentence.
“Increased levels of BMI” – it should be “increased BMI”
“partially unclear” – it does not seem to be proper to describe the condition on this way (page 4)
“early detection of endothelial dysfunction in children and adolescents with diabetes is mandatory to prevent or at least delay cardiovascular complications” – it is hard to accept that it is “mandatory” for prevention. The evidence is not so strong and so well accepted to say it is mandatory.
Author Response
Responses to Referee 2 comments:
>> It should be clearly underlined if there are important differences in the effect of diabetes on CV risk between adults and children/adolescents as well as in treatment strategy.
Authors’ response: We thank the Reviewer for this comment. We enriched our manuscript by introducing the suggested reports. Please, see edited text at page 2, 5 and 6 of the manuscript.
- The manuscript should be corrected by a native English speaker. Editorial work is also needed.
Authors’ response: As suggested our paper underwent an extensive English editing.
- What is T1D 2 ??? (page 2)
Authors’ response: We are very sorry for our incorrect typing and we revised it. Please, see edited text at page 2 of the manuscript. The whole manuscript has been carefully rechecked to eliminate any not-spelled acronyms.
- Table 2 – hsPCR ???- what it is ???
Authors’ response: We are very sorry for our incorrect typing and we revised it. Please, see edited text of Table 2. The whole manuscript has been carefully rechecked
- It is not clear what kind of condition is called “glucose variability”
Authors’ response: We thank the Reviewer for the valuable comment and we specified it. Please, see edited text at page 3 of the manuscript.
- It dose not to be proper to use “ischemic electrocardiogram” in this sentence.
Authors’ response: We addressed the Reviewer comment by deleting “ischemic electrocardiogram” from our manuscript. Please, see edited text at page 3 of the manuscript.
- “Increased levels of BMI” – it should be “increased BMI”
Authors’ response: We addressed the Reviewer comment by rewriting this expression. Please, see edited text at page 3 of the manuscript.
- “partially unclear” – it does not seem to be proper to describe the condition on this way (page 4)
Authors’ response: We addressed the Reviewer comment by rewriting this expression. Please, see edited text at page 4 of the manuscript.
- “early detection of endothelial dysfunction in children and adolescents with diabetes is mandatory to prevent or at least delay cardiovascular complications” – it is hard to accept that it is “mandatory” for prevention. The evidence is not so strong and so well accepted to say it is mandatory.
Authors’ response: We addressed the Reviewer comment by rewriting this sentence. Please, see edited text at page 4 of the manuscript.

Round 2
Reviewer 2 Report
The manuscript was found significantly omproved. However, still an improvemebt is needed.
page 7" - I agree that it does not make sense: " In the management of children and adolescents with diabetes, a pivotal role is played by therapeutic education"
page 7: " Therapeutic drugs" - do we use non-terapeutic drugs to treat diseses? Meybe we have "terapeutic strategies" on mind
Author Response
Responses to Referee 2 comments:
>> The manuscript was found significantly improved. However, still an improvement is needed.
Authors’ response: We thank the Reviewer for this comment and we improve our manuscript. We also
page 7" - I agree that it does not make sense: " In the management of children and adolescents with diabetes, a pivotal role is played by therapeutic education"
Authors’ response: We addressed the Reviewer comment by rewriting this expression. Please, see edited text at page 7 of the manuscript
Page 7: “Therapeutic drugs” -do we use non-therapeutic drugs to treat disease? Maybe we have “terapeutic strategies” on mind
Authors’ response: We addressed the Reviewer comment by rewriting this expression. Please, see edited text at page 7 of the manuscript
